# Histotripsy of Liver Tumors: Patient Selection, Ethical Discussions, and How We Do It

**DOI:** 10.3390/cancers17071100

**Published:** 2025-03-25

**Authors:** Melis Uysal, Chase J. Wehrle, Sangeeta Satish, Emily Knott, Hanna Hong, Erlind Allkushi, Andrea Schlegel, Eren Berber, Federico Aucejo, JaeKeun Kim, David C. H. Kwon

**Affiliations:** Cleveland Clinic, Department of General Surgery, Digestive Disease & Surgery Institute, Cleveland, OH 44120, USA; uysalm@ccf.org (M.U.); wehrlec@ccf.org (C.J.W.); satishs2@ccf.org (S.S.); schlega4@ccf.org (A.S.); aucejof@ccf.org (F.A.); kimj30@ccf.org (J.K.)

**Keywords:** histotripsy, liver cancer, ablation

## Abstract

Histotripsy is a non-invasive treatment method that uses precisely controlled acoustic cavitation to disrupt tumors. It offers a potential alternative for patients with liver tumors who are not surgical candidates or who have complex tumors. At our institution, histotripsy is utilized for patients with unresectable tumors, those who need downstaging for liver transplantation, those who are not candidates for curative locoregional therapies, or those who prefer a less invasive option. Early studies show histotripsy’s effectiveness in tumor destruction with minimal side effects. This technique provides a valuable treatment option for patients who would otherwise have limited choices due to advanced disease or ineligibility for other treatments.

## 1. Introduction

Liver malignancies, including primary and metastatic tumors, represent a significant global health burden. Colorectal cancer (CRC) alone ranks as the third leading cause of both cancer incidence and cancer-related mortality worldwide [1]. At diagnosis, approximately 25% of patients are diagnosed with stage IV CRC with synchronous metastases, and nearly 50% of all patients will eventually develop liver metastases; 90% will ultimately die due to metastatic disease [2]. Similarly, hepatocellular carcinoma (HCC) and intrahepatic cholangiocarcinoma (ICC) pose substantial challenges, characterized by limited curative options and high mortality rates.

Surgical resection remains the gold standard for the curative treatment of liver tumors, offering 5-year survival rates of up to 50% for colorectal liver metastases (CRLM) and improved outcomes in select HCC and ICC patients [3,4,5]. However, the majority of patients present with unresectable disease—approximately 85% in stage IV CRC, and similarly high proportions in advanced HCC and ICC—necessitating palliative systemic therapy as the primary treatment option. Unfortunately, systemic chemotherapy and targeted therapies provide limited long-term survival benefits, with 5-year overall survival rates often below 10% [6], underscoring the urgent need for alternative therapeutic options.

Locoregional therapies such as radioembolization (TARE), external radiation therapy (SBRT), Hepatic Artery Infusion (HAI) Pump Chemotherapy, and thermal ablation techniques like radiofrequency ablation or microwave ablation have been integral to the management of primary and metastatic liver tumors [7,8,9]. While these approaches are widely practiced and can provide similar therapeutic effects in small lesions [10,11], they are not without limitations, including their invasive nature, substantial radiation exposure, restricted real-time visualization of therapeutic effects, and inconsistent success in achieving reliable local tumor control [12,13,14,15,16,17,18]. These approaches are usually used as palliative alternatives.

Histotripsy is an innovative, non-invasive technique that employs precisely controlled acoustic cavitation to disrupt tumors without using heat or ionizing radiation, allows for the real-time visualization of the tissue effects, and can also be referred to as an image-guided energy-based ablation method [19]. Preclinical studies by Worlikar et al. have highlighted its effectiveness in achieving tumor ablation and reducing both local progression and metastatic spread in rodent models [20,21]. These encouraging findings sparked research interest in human clinical studies. The THERESA trial evaluated the use of histotripsy in eight patients with a total of 11 liver malignancies, including seven CRLMs in five patients. The trial achieved its primary endpoint, demonstrating acute technical success in all 11 tumors treated with histotripsy. Technical success was defined as creating a targeted zone of tissue destruction within the planned volume, as confirmed by MRI one day post-procedure. Importantly, no adverse events were deemed probably or definitely related to the device, underscoring the excellent safety profile of histotripsy [22]. The #HOPE4LIVER trial involved 44 participants with 49 liver tumors, including 10 cases of CRLM. The study achieved a 95% technical success rate in tumor ablation, significantly exceeding the predefined performance goal of 70%. Major procedure-related complications were observed in approximately 7% of cases, which is within acceptable safety margins for liver-directed therapies and is similar to the COLLISION trial in thermal ablation, which also showed a 7% rate of major complications [23,24]. However, most recently, a large multinational experience of histotripsy used in a standard-of-care fashion demonstrated just a 1% rate of overall major complications within 30 days of histotripsy, which compares very favorably to any other LRT approach [25]. Its safety profile has further led to its use in a bridging context before liver transplant for HCC, in which case cirrhotic liver morphology can preclude the safe use of other approaches [26]. These results further highlight histotripsy’s potential as a safe, non-invasive treatment option for liver malignancies. Furthermore, histotripsy preserves collagenous structures, allowing for the treatment of cancers involving major structures such as the bile ducts and portal or hepatic veins [27]. Finally, there is a proposed immune-priming effect of histotripsy mediated in murine models by CD4/CD8+ T cells, which in turn leads to off-target immunologic destruction of tumors that have received no direct or systemic therapy [28].

## 2. Patient Selection

At our institution, all patients with liver tumors are carefully evaluated through a comprehensive, multidisciplinary weekly liver tumor board. This board consists of a diverse team of specialists, including liver transplant surgeons, liver surgical oncologists, interventional radiologists, abdominal radiologists, medical oncologists, and hepatologists. These collaborative discussions allow for a thorough assessment of each patient’s case, ensuring the most appropriate and individualized treatment recommendations.

Histotripsy is considered based on the collective expertise of the tumor board, with the procedure performed at the discretion of the procedural and surgical providers delivering the therapy. Patients are deemed suitable candidates for histotripsy at our institution under the following circumstances:

1. Ineligibility with current surgical or non-surgical treatment: Patients who are not candidates for curative-intent surgical or non-surgical treatment due to significant medical comorbidities or limited liver function that makes it unsafe or impractical may be candidates. Some tumors may be inaccessible or too large to resect safely, especially if they are near vital structures such as large blood vessels or bile ducts. In such cases, both surgery and certain locoregional treatments, like ablation, may be impractical or ineffective. Our center has begun using histotripsy in cases of severe medical comorbidities. Potential benefits include shorter anesthesia time, no requirement to stop perioperative medications, and removing any need for surgical or interventional wound healing. Preliminary studies demonstrate a favorable safety profile, although more in-depth studies in different populations are needed [25].

2. Unresectable disease with palliative intent: Patients with unresectable liver tumors who are not responsive to current systemic or liver-directed therapy may be candidates. In this context, “palliative intent” refers to the goal of disease control, alleviation of symptoms, and improvement of the patient’s quality of life rather than curative treatment.

3. Downstaging or bridging to liver transplantation: Patients with hepatocellular carcinoma who are ineligible for conventional locoregional therapies, such as Yttrium-90 (Y90), due to tumor proximity to major blood vessels and who require tumor downstaging to meet transplantation criteria or as a bridging therapy to reduce tumor progression while awaiting liver transplantation may be candidates. One such example has been published [26]. The decision to use histotripsy as a downstaging strategy is carefully considered during our weekly multidisciplinary tumor board meetings. This approach has been included as a recommended treatment by the United Network for Organ Sharing (UNOS).

This patient-centered and multidisciplinary approach ensures that histotripsy is thoughtfully integrated into treatment protocols, balancing innovation with safety, clinical evidence, and individual patient needs.

## 3. How We Do It

The Edison System received FDA marketing authorization on 6 October 2023 as a focused ultrasound system for non-thermal mechanical tissue ablation. It is indicated for the non-invasive destruction of liver tumors, including unresectable liver tumors, using a non-thermal, mechanical process of focused ultrasound. Notably, the FDA documentation does not specify particular types or sizes of liver tumors suitable for treatment. The authorization encompasses liver tumors in general, including those deemed unresectable, and does not restrict its use solely to patients without established treatment options. In summary, the Edison System is FDA-authorized for the non-invasive treatment of liver tumors, including unresectable ones, without specific limitations on tumor type or size or prior treatment options.

The equipment used for histotripsy in our center was purchased with institutional approval, and the treatment is provided as part of a standard care protocol. While the manufacturer, HistoSonics, provides clinical support, as do many medical device companies, there is no funding bias involved in the treatment process. All patients are offered the opportunity to participate in clinical research if they consent, and the data collected are intended to contribute to a broader understanding of histotripsy’s efficacy and safety. We remain committed to transparent and unbiased reporting of outcomes, and our clinical practices are driven by the best interests of the patients and scientific integrity.

### 3.1. Pre-Treatment Protocol

Patients are referred for histotripsy as previously described. They are initially reevaluated to determine whether currently established surgical or non-surgical treatment options are available. A detailed description of the advantages and disadvantages of histotripsy is given to the patient before consideration. Our patient discussions are grounded in the available clinical data and tailored to each individual’s diagnosis. While the number of patients treated with histotripsy remains limited, we ensure that patients are informed based on both our direct clinical experience and broader oncologic principles. Patients are provided with transparent information regarding the existing evidence, including the potential benefits, risks, and gaps in knowledge, allowing them to make informed decisions. We are currently working on a manuscript to further detail our patient information and outcomes data, which will contribute to the evolving evidence base for histotripsy.

Once the patient decides to proceed with histotripsy, candidacy is first assessed using this cross-sectional imaging (Figure 1). At present, the application of histotripsy is limited by bone, bowel gas, and depth, which makes certain anatomic locations potentially prohibitive (Figure 2). The subcostal approach is easier to target, but the area it can cover is limited, and it can be expanded using a trans-costal approach. However, most commonly, the very superior lesions in the dome of the liver or deep lesions in the caudate lobe are difficult to target and may not be considered candidates. A crude guide to lesions that are anatomically favorable, intermediate, or unlikely to be feasible is provided (Figure 1). Ultrasound is performed to assess eligibility for histotripsy. As the HistoSonics Edison System uses ultrasound for histotripsy, tumors that are undetectable by the ultrasound probe cannot be treated using this technique. Fusion technology is available within the current platform, though our center does not treat images that cannot be visualized on ultrasound. The technology developer is actively in the process of bringing a cone-beam CT-based platform to market, although we cannot yet comment definitively on the timeline or reliability of this approach. The visibility, echogenicity, depth, method of approach (subcostal vs. trans-costal), and angle of approach are described in detail by the ultrasonographer to evaluate the likelihood of successful treatment and are graded accordingly. Dedicated ultrasonography to evaluate candidacy for histotripsy helps rule out tumors that are not targetable, but it does not mean that tumors seen on ultrasonography will be definitively treatable since the acoustic window of histotripsy does not completely align with the window observed during ultrasound evaluation. This is a limitation of the technology, but we believe the relatively minor nature of the pre-operative ultrasound is helpful, and in some cases, patients are ruled out when ultrasonographers believe that the tumor is definitely not targetable. We have developed a standard template for the best communication between teams and to obtain objective data for future studies. Once eligibility is confirmed, patients proceed with a comprehensive pre-procedure workup designed to establish baseline assessments for accurate monitoring of treatment outcomes. This thorough evaluation focuses on tumor characteristics and overall health to optimize treatment planning.

#### 3.1.1. Laboratory Tests

Patients undergo a complete blood count (CBC) and a comprehensive metabolic panel (CMP) to assess their overall health and liver function. Tumor markers, such as carcinoembryonic antigen (CEA) for colorectal liver metastasis, CA19-9 for cholangiocarcinoma, CA-125 for ovarian cancer, and AFP for hepatocellular carcinoma, are measured to provide additional information.

#### 3.1.2. Circulating Tumor DNA (ctDNA)

A blood sample is collected to analyze ctDNA, a minimally invasive biomarker that can detect genetic alterations and tumor burden. ctDNA levels offer valuable insight into the molecular profile of the tumor and can serve as an early indicator of treatment response or recurrence. ctDNA has been well-described by our group in the management of CRLM and other liver malignancies, including pre-operative selection and postoperative surveillance [29,30,31,32,33]. While novel, we hope that ctDNA will serve in a similar capacity in relation to histotripsy.

### 3.2. Procedure

Histotripsy is performed using the HistoSonics Edison System (HistoSonics, Plymouth, MN, USA) under general anesthesia in an operating room setting (Figure 3). The Edison System used in our clinic is a non-invasive extracorporeal focused ultrasound device. It consists of a robotic arm-mounted transducer that allows for precise targeting of the lesion. The treatment is guided by real-time ultrasound imaging, which enables the physician to visualize cavitation bubble activity and dynamically monitor treatment progress. This ensures accurate energy delivery and allows for adjustments if necessary. The precise location and dimensions of the tumor are mapped with the HistoSonics system, and targeting algorithms in the system optimize the delivery of focused ultrasound to the defined tumor region.

Histotripsy has been explored for the treatment of liver tumors, including those that are unresectable. The size of lesions that can be effectively treated depends on various factors, including the device’s specifications and the tumor’s characteristics. While specific size limitations are not universally defined, ongoing clinical trials aim to establish the optimal parameters for treating different lesion sizes. In our experience, lesions up to 4 cm have been successfully treated during a single treatment application of the device. However, larger lesions are targetable using multiple overlapping treatment zones, although we have only used this method ourselves in a very limited number of cases. Importantly, histotripsy operates through mechanical disruption, avoiding thermal effects during energy transfer to the tissue. This non-thermal mechanism has been documented in preclinical and clinical studies, demonstrating effective tissue ablation without heat-induced damage [34].

The whole procedure lasts approximately 2–4 h, of which 10–45 min are expected to be actual treatment time. Patients are initially positioned supine on the table to perform ultrasonography and obtain pre-procedure imaging, allowing optimal access to the target tumor. Trans-costal treatments are most frequently performed in a “lazy lateral” position, while sub-costal treatments are performed supine.

Lesion targeting and the selection of anatomically favorable lesions are critical. Figure 2 includes an image-based guide for lesions that are typically favorable (green: Segments 1, 2, 4a, 6), those that are possibly favorable with advanced targeting experience (e.g., trans-costal approach: Seg 3, 4B, 5), and those that are less likely to be favorable with current ultrasound-based techniques (red: Seg 7, 8, some 4A).

### 3.3. Post-Treatment Follow-Up

Patients undergoing histotripsy are typically discharged on the same day as the procedure, reflecting the minimally invasive nature and low risk of complications associated with the treatment. A structured follow-up schedule is implemented to closely monitor recovery, assess treatment efficacy, and guide further clinical decisions.

If direct treatment is applied to major vascular structures, patients are initiated on therapeutic anticoagulation, usually with direct oral anticoagulation (DOACs), for two weeks after treatment. This treatment is initiated due to growing evidence of transient thrombosis after other ablation modalities, such as thermal ablation [35,36,37].

#### 3.3.1. POD 1

Patients return for outpatient imaging and laboratory testing on POD1. A tri-phasic CT scan of the liver is performed to evaluate the immediate effects of histotripsy, including tumor disruption and any potential complications such as thrombosis, ascites, etc. Recently presented international work demonstrates that such complications are very rare, but due to the novel nature of the procedure, extra caution is currently being taken. Bloodwork includes standard postoperative labs, such as a CBC, CMP, and serum tumor marker levels, to assess overall health and detect any abnormalities.

To gain a clearer understanding of the short- and long-term effects of histotripsy, our team established a standardized imaging time point at 24 h post-treatment. This approach allows us to assess immediate post-treatment changes while also providing a consistent reference for evaluating longer-term outcomes. By ensuring uniform data collection, we can better track the evolution of histotripsy-induced tissue effects and optimize its clinical application.

#### 3.3.2. POD 14

A virtual follow-up appointment was scheduled to review the results of initial testing and assess the patient’s recovery. Providers discuss any symptoms, address patient concerns, and evaluate the progress of healing remotely, minimizing the need for unnecessary in-person visits. These visits are now considered optional as our experience has grown and are conducted according to patient and physician preference, based on individual case characteristics.

#### 3.3.3. POD 30

Patients return for an in-person evaluation to ensure complete recovery and assess the longer-term effects of the procedure. Before this visit, patients undergo a comprehensive workup, including a tri-phasic CT scan of the liver to measure changes in tumor size and morphology, CBC, CMP, and tumor marker levels, as well as ctDNA testing to evaluate tumor burden and detect potential residual disease. These results are reviewed with the surgeon, allowing for tailored recommendations on ongoing care.

#### 3.3.4. POD 90

Patients undergo a repeat evaluation with the same tests, including imaging and blood work. The results are discussed during the patient visit. The 90-day visit is also unique in that the potential for re-treatment is considered at this time.

### 3.4. Management of Concomitant Therapies

To date, there are no documented interactions between histotripsy performed using the HistoSonics Edison^®^ System and concomitant pharmacologic treatments for liver malignancies. This unique non-invasive technology, which relies on acoustic cavitation rather than thermal, ionizing, or chemical mechanisms, operates independently of the metabolic or pharmacodynamic pathways that might be influenced by systemic therapies such as chemotherapy or targeted agents. The absence of interaction concerns allows histotripsy to complement ongoing systemic therapies. This synergistic approach can maximize therapeutic efficacy, particularly in patients with advanced or multifocal disease, without compromising treatment continuity. Thus, in short, we continue all scheduled systemic medications throughout the peri-procedural period, including chemo/immunotherapy and, importantly, all anticoagulation. We note that anticoagulation is continued, including on the morning of surgery, without interruption. We have not seen a bleeding event to date.

Patients undergoing histotripsy are ensured access to the full spectrum of supportive care to address procedural needs, manage potential side effects, and maintain overall well-being during treatment.

## 4. Discussion and Ethical Considerations

This is the first description of an institutional approach to a radically novel technology recently FDA-cleared for the treatment of liver tumors. In our recent analysis, 18 hospitals have treated patients with this technology, and >50 additional hospitals are awaiting the technology. This makes the adoption of clinical protocols of immediate programmatic relevance. Such standardization will also aid research and further scientific advancement by allowing for cross-institutional comparison.

The use of histotripsy presents some challenges due to the limited clinical evidence currently available. As a new technology, its use requires a careful balance between exploring its full potential and maintaining rigorous standards for patient safety and ethical care. The use of histotripsy is justified by FDA clearance for liver tumors, but its application requires more thoughtful consideration since its indication is supported by very little evidence and is not clearly defined. When considering histotripsy, it is crucial to compare it to alternative interventions that have a stronger foundation of clinical data. The effectiveness, safety profiles, and long-term outcomes of established locoregional therapies, including thermal ablation, chemoembolization, and surgical resection, are well-supported by extensive clinical evidence and documented research [38,39,40,41,42,43]. While these methods are effective, they can be invasive, have higher complication rates, and have limitations in certain patient populations. On the other hand, histotripsy provides a targeted, non-invasive method with promising initial outcomes, including minimal complications and short recovery periods. Clinicians must, however, balance the potential advantages of histotripsy against the unpredictability that comes with a novel treatment method. The justification for histotripsy lies in its potential to address unmet needs in current treatment options, such as for patients who are not surgical candidates or who seek less invasive treatment options. While histotripsy has admittedly fewer data than other interventions, there are many populations in which it seems to offer immediate clinical advantages and an improved safety profile, such that its immediate use might benefit patients in the critical setting.

A fundamental component of ethical medical practice is the “do no harm” principle. When compared to invasive procedures like thermal ablation or surgical resection, histotripsy has shown an exceptional safety profile with noticeably lower rates of complications [25]. Its ethical use is supported by its low-risk profile, especially for patients who might not be able to tolerate conventional treatment methods because of comorbidities or advanced disease. Histotripsy is also being utilized in palliative settings with the goal of inducing off-target immunologic tumor destruction. However, further data are needed to refine patient selection and optimize its application. Given its demonstrated safety, patient tolerance, and the ability to continue systemic therapies uninterrupted, we have adopted a relatively broad clinical approach while awaiting more supporting evidence. As a non-invasive ablation technique, histotripsy has shown promising results in our clinic, suggesting a potential synergistic effect that warrants further investigation. It is imperative that the patient is fully aware of the limitations of histotripsy and that data regarding oncological outcomes are very limited before making a decision.

A case report from our group recently described a patient who underwent liver transplantation following histotripsy in our clinic. It documents the groundbreaking use of histotripsy as a bridging therapy prior to liver transplantation for hepatocellular carcinoma (HCC), marking a significant step in advancing treatment modalities [26]. The findings provide post-trial evidence of a complete pathological response, demonstrated by total necrosis of the treated lesion observed in the explanted liver. Despite histotripsy’s non-invasive nature and the lack of histological data from previous treatments, liver transplantation offered a unique opportunity to confirm the therapy’s effect. This allows for a potential de-coupling of the oncologic urgency of transplant from the management of liver disease. Despite the increasing availability of grafts with machine perfusion, transplant oncology continues to represent a major disease burden, and studies have already shown how other LRTs can impact post-transplant outcomes [44,45,46,47].

It is critical to note that at present, data on histotripsy has not come close to the level available for other indications, such as Y90, thermal ablation, or surgery [11,24,48,49]. Thus, we cannot recommend the use of histotripsy at present when another method is clearly indicated. However, we believe there are some indications, as described, in which other protocols are not as commonly indicated and case histotripsy can fill gaps. Furthermore, the safety profile as cited presents an exciting opportunity to innovate the future of cancer care in a relatively safe manner [25]. Medium- and long-term oncologic data are critical for establishing the optimal use cases for this technology. In this vein, our center is actively designing two trials: one using histotripsy as an adjunctive therapy for unresectable advanced CRLM in addition to first-line platinum-based chemotherapy and another using histotripsy as a bridging therapy prior to liver transplantation for HCC. The latter offers potential additional benefits in obtaining explanted tissue for histopathology.

These human results align with preclinical studies by Worlikar et al., which showed complete tumor regression using histotripsy in animal models [20]. The study’s outcomes are particularly promising, showing complete tumor necrosis at the cellular level. This finding raises the possibility of histotripsy serving as a curative treatment, as suggested by preclinical evidence. However, in the described case, cirrhosis due to metabolic-associated steatotic liver disease (MASLD) necessitated liver transplantation. Without the presence of cirrhosis, histotripsy might have provided a standalone curative solution. For this patient, conventional bridging therapies were not feasible due to hepatic–pulmonary arterial shunting. Histotripsy became the selected treatment, highlighting its potential to address gaps in current therapeutic options. The success of histotripsy in this case underscores its potential as a valuable addition to HCC treatment strategies, particularly as a non-invasive alternative to more risky and invasive bridging methods.

## 5. Conclusions

Liver tumors present significant challenges in cancer treatment, with limited options for patients deemed ineligible for surgical resection. Histotripsy is emerging as an innovative and non-invasive approach with the potential to address these gaps, offering precise tumor destruction with minimal damage to surrounding tissue. Herein, we describe our institutional approach to histotripsy before, during, and after the procedure in an attempt to inform this growing practice. While more research is needed to confirm long-term benefits, histotripsy provides a promising alternative for managing liver tumors and advancing liver cancer care.

## Figures and Tables

**Figure 1 cancers-17-01100-f001:**
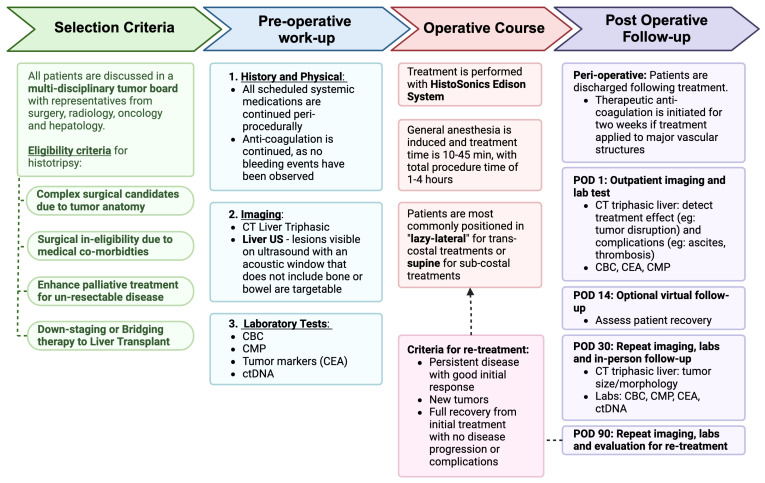
Histotripsy treatment protocol within our institution for colorectal cancer liver metastasis. Created in https://BioRender.com, accessed on 15 February 2025.

**Figure 2 cancers-17-01100-f002:**
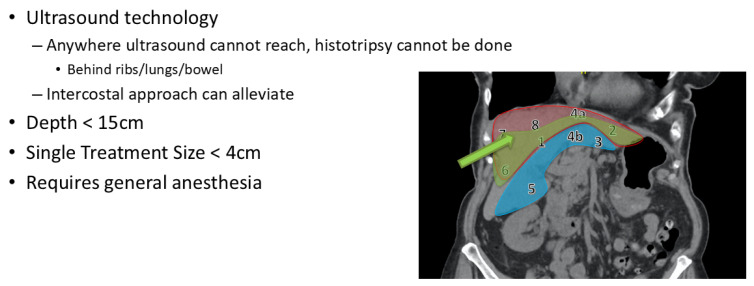
Histotripsy treatment limitations and guide to anatomic locations: likely (green, optimal arrow location), moderately likely (blue), or unlikely to result in technical success (red). Numbers indicate liver segments.

**Figure 3 cancers-17-01100-f003:**
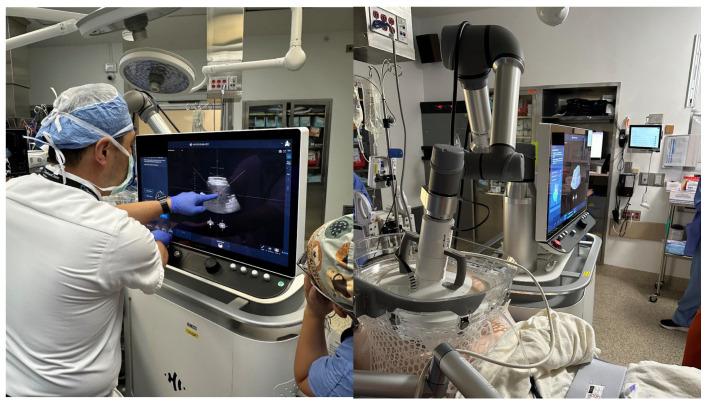
Histotripsy being performed using the HistoSonics Edison System (HistoSonics, Plymouth, MN, USA) under general anesthesia in an operating room setting.

## Data Availability

Data sharing is not applicable to this article as no datasets were generated or analyzed during the current study.

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
