# Peer review of "Histotripsy of Liver Tumors: Patient Selection, Ethical Discussions, and How We Do It"

_cancers, 2025, doi:10.3390/cancers17071100_

Round 1
Reviewer 1 Report
Comments and Suggestions for Authors
The manuscript describes a new non-invasive treatment method and its application in the treatment of liver tumors.
General comments:
Regarding the treatment procedure, a more detailed description would be welcome. Is the device hand-held? Do the performing doctor get a live feedback/image on the progression of the treatment? How is delivery of energy to the target assured? A figure/photo of the device and a treatment session would perhaps be helpful.
The treatment in liver: How large lesions can be treated? Is there a limitation regarding size? No thermal effect at all when transferring energy to a tissue? How is this documented?
A detailed description of institution routines, including follow-up, is given. It would be of great interest to know how many patients that have been treated with the method described in the author’s own institution, over how long time, and short-time results (or long-term if available) regarding both LTP and complications for these treatments. After all, the method itself has been previously described, and the paper in its current form adds little new except for presenting local routines.
Specific comments:
Abstract.
L 19. “ .. and required interruption of systemic treatments..”. Is this really required with thermal ablation? Do the authors have a reference for this?
Introduction
P1, l 37.
“Surgical resection of CRLM remains… “
This view has been challenged in a recent paper in Lancet Oncology, Jan -25 reporting the results from a RCT (COLLISION-trial) and also in the Maverrick study.
P2, l 44-50.
When describing locoregional therapies, recent references such as the above mentioned references should be mentioned, in my opinion. Regarding reliable local tumor control with thermal ablation recent literature describing improved control with ablation conformation exist, and references from 2006/2008 do not necessarily reflect neither current nor best practice.
P 2, l 66-67:
“Major… complication.. appr. 7 %”.
This is on the same level as for thermal ablation, e.g., 7% in the COLLISION trial. Whether Histotripsy really offers an advantage, regarding complications, compared to thermal ablation is thus unclear. Unlessens the authors can report superior results in their own institution this point would be appropriate to mention in the discussion.
P2, l70:
“..histotripsy preserves collagenous structures….. treatment of major structures such as the portal or hepatic veins”
What about bile ducts? Do they “survive”? This is of great interest as vessels is mostly not an issue with thermal ablation, while central bile ducts are.
Conclusions
P7, l 275-282
“CRLM.. limited options for patients deemed ineligible for .., resection….histotripsy promising alternative…. Long-term benefit”
The basis for the conclusions is unclear.
Complications with the method seem to be on the same level as for thermal ablation. Whether it gives an advantage compared to thermal ablation regarding central tumors is unclear and data regarding local tumorcontrol limited in short-term as well as long-term. To me it is not clear witch “gaps2 the method addresses that is not already covered by thermal ablation and//or IRE. Not using a needle is of course appealing, but if it constitute an advantage is unclear.
Reviewer 2 Report
Comments and Suggestions for Authors
In the present manuscript author try to discuss novel concept of Histotripsy of Liver Tumors. However overall manuscript flow is interesting. But few major flaws are hindering the outcome. Such as
The rationality of these type of studies are not discussed properly.
Data authenticity compared with other methods are missing.
Further management of critically ill patients is missing. How it is useful toward aged patients.
Reviewer 3 Report
Comments and Suggestions for Authors
Histotripsy of Liver Tumors: Patient Selection, Ethical Discussions and How We Do It
Simple Summary:
- Should clarify that the method is offered to patients that are not candidates for resection and/or standard of care thermal ablation for Colorectal liver metastases.
The “limited choices” need be clarified. NCCN Guidelines recommend thermal ablation as a stand-alone treatment or in combination with surgery as a local curative treatment for selected small CLM. Offering Histotripsy to this population is not supported by the current level of evidnce and is outside the standard of care and the NCCN guidelines.
Abstract
Need Revision according to points below:
- A clarification on the use of local and ocoregional therapies is needed. Ablation is offered as local cure instead of or in combination with surgery top eradicated disease.
- Chemoembolization is offered in advanced disease that is not amenable to local cure.
- Radioembolization is also an option for CLM.
- Image guided radiotherapy is also an option with less data to support is use.
- Irreversible Electroporation is also an option for high risk patients similarly to histoptripsy.
- The term “palliative” is a misnomer and should be clarified in local and locoregional use.
Introduction.
- The statements about treatments other than surgery are summed up as equal which does not reflect the level of evidence. Recent RCT trial has indicated that Thermal ablation was similar to resection in terms of survival and progression free survival and it was better than surgery with regards to per tumor local control and safety. A complete revison is required after reviewing and discussing relevant publications.
- Similarly to prior comments, the statements here need reflect the NCCN recommendation as in the evidence blocks, Principles of surgery and locoregional therapy for metastatic colorectal disease.
- The nice summary of histotripsy outcomes here clearly indicates the premature phase to even propose this treatment as an alternative to standard of care thermal ablation. The data for histotripsy clearly lack the evidence for other local therapies as well. Specifically, IRE, Radiation segmentectomy and even Image guided radiotherapy. This needs be clearly stated. Regarding the immunomodulation effects in humans there is no evednec to support this for histotripsy, thus such approach could be only supported via a clinical trial.
- Patient Selection.
- Please clarify the ineligibility for non-surgical treatments
- What is the palliative intent? Does this mean disease control, understanding that there will be no cure? And if so please elaborate
- Is Histotripsy offered instead of thermal ablation IMRT or Y90 as a bridge to transplant? And if so, what are the data supporting this approach?
3 How we Do it
- Are tumors that are not detected by ultrasound absolutely ineligible for histotripsy? Could fusion methodology be offered?
3.3. Please provide the evidence regarding treatment related thrombosis
3.3.1: why is a triphasic examination performed 24 hours after and not on the same day of histotripsy?
3.4. Please provide references for any synergistic effects of histotripsy with chemotherapy. Otherwise please remove these statements.
- Discussion and Ethical Considerations
- Histotripsy is considered an ablative treatment. It should be practiced according to guidelines of image guided ablation. We recommend adherence to the reporting standards for ablation as described by Ahmed M et al; Radiology 2014 to allow similar terminology and comparisons.
- The data for histotripsy are minimal and do not justify its use in patiens that are eligible for curative intent thermal ablation with margins. This cannot be justified by the “do no harm approach” and needs further discussion.
- The statement that Histotripsy has a better safety profile than image guide thermal ablation is not that clear. With 7% complication rate for histotripsy and the incidence of venous thrombosis this statement needs be revised accordingly.
- Thermal Ablation has been safe near vessels without increased incidence of thrombosis. How does this compare to Histotripsy?
- The statement of “Palliative” treatment need be removed. Simliarly the notion that there is immunomodulation creates unsubstantiated hope and it is truly anethical. It should be clear that this approach must be done only within a well-designed clinical trial.
- Lines 262-273 do not reflect the current level of evidence for histotripsy and need be completely revised. These lines are more of a description of the probabilities and promise of this treatment and not the current status of affairs.
- Also the discussion goes back and forth HCC and CLM. Need focus to one pathology or revise the entire paper to more universal review of How to do it removing statements that are not supported by existing evidence.
- Thermal ablation success strongly relies in the ability to create margins similarly to resection. Has this been assessed or considered for when performing Histotrispy? Please discuss.
References
An extensive list of relevant references especially regarding ablation of Colorectal liver metastases is missing and need be reviewed and discussed if this review is focusing in this disease.
Below is a short and very relevant list that need be discussed.
- Ahmed M, International Working Group on Image-Guided Tumor Ablation; Interventional Oncology Sans Frontières Expert Panel; Technology Assessment Committee of the Society of Interventional Radiology; Standard of Practice Committee of the Cardiovascular and Interventional Radiological Society of Europe. Image-guided tumor ablation: standardization of terminology and reporting criteria-a 10-year update. J Vasc Interv Radiol. 2014 Nov;25(11):1691-705.e4. doi: 10.1016/j.jvir.2014.08.027. Epub 2014 Oct 23. PMID: 25442132; PMCID: PMC7660986.
- Benson AB, Colon Cancer, Version 3.2024, NCCN Clinical Practice Guidelines in Oncology. J Natl Compr Canc Netw. 2024 Jun;22(2 D):e240029. doi: 10.6004/jnccn.2024.0029. PMID: 38862008.
- Tinguely P, Ruiter SJS, Engstrand J, de Haas RJ, Nilsson H, Candinas D, de Jong KP, Freedman J. A prospective multicentre trial on survival after Microwave Ablation VErsus Resection for Resectable Colorectal liver metastases (MAVERRIC). Eur J Cancer. 2023 Jul;187:65-76. doi: 10.1016/j.ejca.2023.03.038. Epub 2023 Apr 5. PMID: 37119639.
- van der Lei S, , Zonderhuis BM, Swijnenburg RJ, van den Tol MP, Meijerink MR etal: Thermal ablation versus surgical resection of small-size colorectal liver metastases (COLLISION): an international, randomised, controlled, phase 3 non-inferiority trial. Lancet Oncol. 2025 Jan 20:S1470-2045(24)00660-0. doi: 10.1016/S1470-2045(24)00660-0. Epub ahead of print. PMID: 39848272.
- Vasiniotis Kamarinos N, et al: Biopsy and Margins Optimize Outcomes after Thermal Ablation of Colorectal Liver Metastases. Cancers (Basel). 2022 Jan 29;14(3):693. doi: 10.3390/cancers14030693. PMID: 35158963; PMCID: PMC8833800.
- Vasiniotis Kamarinos N et al: 3D margin assessment predicts local tumor progression after ablation of colorectal cancer liver metastases. Int J Hyperthermia. 2022;39(1):880-887. doi: 10.1080/02656736.2022.2055795. PMID: 35848428; PMCID: PMC9442248.
- Laimer G, et al: Multicenter and inter-software evaluation of ablative margins after thermal ablation of colorectal liver metastases. Eur Radiol. 2024 Aug 2. doi: 10.1007/s00330-024-10956-5. Epub ahead of print. PMID: 39093415.
- Zirakchian Zadeh M, Sotirchos VS, Kirov A, Lafontaine D, Gönen M, Yeh R, Kunin H, Petre EN, Kitsel Y, Elsayed M, Solomon SB, Erinjeri JP, Schwartz LH, Sofocleous CT. Three-Dimensional Margin as a Predictor of Local Tumor Progression after Microwave Ablation: Intraprocedural versus 4-8-Week Postablation Assessment. J Vasc Interv Radiol. 2024 Apr;35(4):523-532.e1. doi: 10.1016/j.jvir.2024.01.001. Epub 2024 Jan 10. PMID: 38215818.
- Chlorogiannis DD,et al. The Importance of Optimal Thermal Ablation Margins in Colorectal Liver Metastases: A Systematic Review and Meta-Analysis of 21 Studies. Cancers (Basel). 2023 Dec 12;15(24):5806. doi: 10.3390/cancers15245806. PMID: 38136351; PMCID: PMC10741591
- Shady W., et al. Percutaneous Microwave versus Radiofrequency Ablation of Colorectal Liver Metastases: Ablation with Clear Margins (A0) Provides the Best Local Tumor Control. J. Vasc. Interv. Radiol. 2018;29:268–275.e1. doi: 10.1016/j.jvir.2017.08.021.
- Izaaryene J. et al: Computed tomography-guided microwave ablation of perivascular liver metastases from colorectal cancer: A study of the ablation zone, feasibility, and safety. Int. J. Hyperth. 2021;38:887–899. doi: 10.1080/02656736.2021.1912413
- Lin Y.-M. et al: Ablative Margins of Colorectal Liver Metastases Using Deformable CT Image Registration and Autosegmentation. Radiology. 2023;307:e221373. doi: 10.1148/radiol.221373.
Reviewer 4 Report
Comments and Suggestions for Authors
Histotripsy of Liver Tumors: Patient Selection, Ethical Discussions and How We Do It
Melis Uysal et al
Abstract
Mainly background of CRC met treatment. Mention of possible target group for histotripsy. Aim of review appears to be discussion of Institutional protocols.
Introduction
1. Informative discussion on current therapy responses.
2. Two prior cohort studies mentioned (Theresa and Hope4Liver)
Included mixed cancer types. What was data specifically for CRC mets?
3. Nature of early phase trials unclear. Were they academic led or commercially sponsored?
4. “Allowing direct treatment of major structures such as the portal or hepatic veins[21]”.
a. I think this should be “allowing treatment of cancers involving major structures”
5. Above data very preliminary. Should mention current status of the device eg licensed/ CE marked? FDA approved? MHRA approved? EMA approved? for therapy and under what circumstances.
Methods
Patient selection :
Suitable patients discussed at tumour board.
1. Is “liver surgical oncologist” a specialist in liver resection?
2. Given that SBRT is a treatment option no radiotherapy input.
3. Large group. How are decisions agreed?
4. What is governance structure to ensure tumour board decisions valid?.
5. Downstaging for liver transplant suggested but applies to HCC and CRC mets topic of review. Review does not include data on HCC management.
How we do it.
1. It is suggested
“Detailed description of the advantages and disadvantages of histotripsy is given to the patient before consideration”. How is this possible with such limited background data, particularly with CRC mets?
2. Suggestion that right dome and caudate lobe lesion not possible to treat. Evidence for this not provided.
3. US assessment mentioned but if poorly correlated with acoustic window is this of value?
4. Pre-procedure work up mentioned, should identify this is in Figure 1.
5. Staging in work up does not include chest CT or PET. Does this suggest that liver directed therapy would be considered independent of presence and progression of other site disease?
6. Laboratory tests for other cancer types included. Should focus article on CRC mets.
7. US evaluation done under GA in operating room.
Limited detail of anaesthetic requirement. Presumably with neuromuscular blockade. Is limited ventilation interruption required?. Why not performed prior to admission for treatment?. Are there no findings that would suggest histotripsy should not proceed.?
8. Limitations and technical considerations should be identified as an experience based protocol rather than an evidence based guidance.
9. Post treatment anticoagulation. The data to support growing evidence of transient thrombosis not provided.
10. Follow up imaging focused only on liver and not assessing possible disease progression elsewhere. How will this “allow for tailored recommendations on ongoing care”
Results : No data from reviewing institute provided or comparison of the risks and efficacy of different Institutional protocols as suggested to be aim of the review in abstract.
Discussion
1. Standardisation of protocols between institutions would seem to be an important aim. Given the lack of data from sites the method of achieving this is unclear. The authors should discuss the published pathways for device development and whether these are applicable.
2. The lack of evidence is acknowledged by authors. Comparison with alternative technologies is suggested but which would be most appropriate and the possible end point of the study has not been addressed. Recruitment to studies comparing surgical resection with non surgical therapies notoriously difficult and the practicalities should be acknowledged.
3. The authors suggest “Histotripsy is also used in a palliative setting….” Implying that it is a curative therapy but without providing evidence to support this comment. Anecdote is not evidence.
4. The use in downstaging of HCC is of interest but if the purpose of this review is to discuss the management of CRC mets then this is not relevant.
Round 2
Reviewer 1 Report
Comments and Suggestions for Authors
Improved after revision. Acceptable for publication.
Reviewer 2 Report
Comments and Suggestions for Authors
It can go further publication procedure.
Comments on the Quality of English LanguageNA
Reviewer 4 Report
Comments and Suggestions for Authors
Thanks for revisions
A few issues not completely addressed
- No mention of current status of the system and its use. FDA approved but no indication of scope of use. What type and size of liver cancers? Is it only licensed for those who have no established treatment options?
2. No response to patient information. Limited numbers of CRC patients treated. Is outcome information given to patients extrapolated from all cancers treated or are CRC met patients provided with limited experience on treating CRC mets?
3. Being used as second line therapy. Are patients being consented as a trial therapy under a research protocol?
3. Possible funding bias not addressed. Was the equipment purchased and treatment being provided on institutional approval? Or was the equipment provided by manufacturer to obtain clinical outcome data ?
Transparency is essential to prevent sources of bias.
